# The Prevalence of *Candida albicans* and *Malassezia globosa* in Preschool Children with Severe Early Childhood Caries: A Case-Control Study

**DOI:** 10.3390/healthcare12131359

**Published:** 2024-07-08

**Authors:** Vanessa C. W. Man, Prasanna Neelakantan, Cynthia K. Y. Yiu

**Affiliations:** 1Paediatric Dentistry, Faculty of Dentistry, The University of Hong Kong, Hong Kong; mvanessa@connect.hku.hk; 2Arthur A. Dugoni School of Dentistry, University of the Pacific, San Francisco, CA 94103, USA; pneelakantan@pacific.edu

**Keywords:** early childhood caries, children, *Candida albicans*, *Malassezia globose*, saliva, plaque

## Abstract

**Background:** This cross-sectional study aimed to identify the prevalence of *Candida albicans* and *Malassezia globosa* in children with severe early childhood caries and caries-free children in Hong Kong. **Methods:** This study first recruited a total of 80 children aged between 48 and 72 months old, 40 children with severe early childhood caries, and 40 caries-free children. The children were then further divided into four groups, with 20 children in each group: Group 1: Severe early childhood caries—*C. albicans*, Group 2: Severe early childhood caries—*M. globosa*, Group 3: Caries-free—*C. albicans* and Group 4: Caries-free—*M. globosa*. Saliva, plaque, and caries lesion samples were collected from participants with severe early childhood caries, while only saliva and plaque samples were collected from caries-free participants. Caries status of the primary molars was assessed using WHO’s decayed, missing, and filled tooth index, and the severity of cavitated lesions was determined based on International Caries Diagnosis and Assessment System criteria as caries code 5 or 6. The samples were analyzed using an Internal Transcribed Space and Quantitative Real-Time Polymerase Chain Reaction. **Results:**
*C. albicans* was more prevalent in saliva and plaque samples of severe early childhood caries than in the caries-free group. Proportion of *C. albicans* in both saliva and plaque samples differed significantly between severe early childhood caries and caries-free groups (*p* < 0.05). Within the severe early childhood caries group, the proportion of children with *C. albicans* varied between 6 and 46%. No significant difference in *M. globosa* load was found between plaque samples of the severe early childhood caries and caries-free groups (*p* = 0.159). Conversely, no significant difference in *M. globosa* load was observed between saliva samples of severe early childhood caries and caries-free groups (*p* = 0.051). **Conclusions:** This study demonstrated a strong association between *C. albicans* and severe early childhood caries. *M. globosa* was detected in both the caries-free and severe early childhood caries groups, albeit at low levels.

## 1. Introduction

Early childhood caries (ECC) poses a significant global public health challenge. According to a recent report by the World Health Organization (WHO), dental caries affects 60–90% of children worldwide [1]. Early childhood caries, a specific type of caries that affects preschool children, is characterized by the presence of one or more decayed lesions and missing or filled tooth surfaces in any primary tooth in children younger than 71 months of age [2]. Severe early childhood caries (S-ECC) is recognized as a particularly aggressive and difficult-to-treat variant of this disease [1,3]. Previous studies have shown that over 90% of decayed teeth in affected children go untreated [4], resulting in extensive damage to the tooth structure, severe pain, tooth loss, and a diminished quality of life [5]. It is worth noting that more than 50% of preschool children in Hong Kong suffer from dental caries [4]. 

Recent in vivo and in vitro studies have suggested that yeasts, mainly *Candida*, co-colonize with *Streptococci* to form carious lesions, especially in cases of S-ECC [6,7,8]. This cross-kingdom synergism between *Streptococci* and *Candida* leads to enhanced biofilm development, facilitated by the simultaneous up-regulation/activation of genes related to biofilm matrix production and carbohydrate metabolism. As a result, this creates a hypervirulent ecological niche that contributes to the pathogenesis of S-ECC [6,9]. In a landmark study, Xiao et al. [6] used salivary and plaque analysis to propose that the presence of *Candida* species, specifically *C. albicans*, results in the formation of a highly acidogenic and acid-tolerant bacterial community in S-ECC. Therefore, *Candida* species are hypothesized to be equally significant contributors to the pathological process, particularly in cases of S-ECC [9]. *Candida* species possess critical virulence traits that promote the development of S-ECC. *Non-albicans Candida* and *C. albicans* can exist as mono, dual, or mixed infections. They are both acidogenic and aciduric, contributing to enamel demineralization and dentin collagenolysis in deep dentinal lesions [9]. However, the precise role of *Candida* in the pathogenicity of S-ECC has yet to be clarified. While some studies have shown no significant correlation between the presence of *C. albicans* and caries, others have clearly demonstrated the cariogenicity of *C. albicans* in rats fed high sucrose/glucose diets [10]. A recent systematic review and meta-analysis [11] concluded that children with ECC have significantly greater odds of having a *Candida* biome compared to caries-free children, with *C. albicans* being the most predominant species. 

Apart from *Candida*, fungi in the genus *Malassezia* (previously known as *Pityrosporum*) are also associated with human skin diseases, such as atopic dermatitis, seborrheic dermatitis, and dandruff [12]. Malassezia has been identified to consist of multiple species, namely *M. globosa*, *M. furfur*, and *M. pachydermatis*, based on their characterization of ribosomal DNA and their ability to grow in different media [13]. A previous study has examined the supragingival mycobiome in dental caries and found that not only was *M. globosa* more abundant in caries-free subjects, but it also outnumbered *Candida* [7]. Moreover, it has been suggested that *M. globosa* has an inhibitory effect against *Streptococcus mutans* and inhibits biofilm formation by *Staphylococcus aureus* via a secreted aspartyl protease [7].

The emerging evidence presented above clearly indicates the potential importance of the mycobiome, specifically the *Candida* biome, in S-ECC. However, there are currently no studies available on the *Candida* biome of S-ECC from Hong Kong. This is clinically relevant, as it is important to understand the strategic intervention approaches required to manage this widespread childhood disease, which could be considered a silent epidemic in Hong Kong and worldwide. Since *Candida* is a prevalent colonizer of ECC [6,9], it is necessary to evaluate its role and the mechanisms by which it contributes to the caries process by assessing its cariogenic attributes [8]. Therefore, this study aimed to compare the prevalence of *Candida*, specifically *Candida albicans* and *Malassezia globosa*, in S-ECC and caries-free (CF) children in Hong Kong. The null hypothesis of this study was that there is no difference in the prevalence of *C. albicans* and *M. globosa* in S-ECC children compared to children without caries. 

## 2. Materials and Methods 

### 2.1. Participant Recruitment and Examination

Prior approval from the HKU/HA HKW Institutional Review Board (UW 20-270) was obtained for this study. Eighty children between the ages of 48 and 72 months, who were attending the Multi-Specialty Clinic of the Institute for Advanced Dentistry, Faculty of Dentistry, the University of Hong Kong, were recruited as participants. Informed consent was obtained from the parents of each child participant. A full dental examination was conducted for all the healthy and co-operative participants by a postgraduate student in Paediatric Dentistry, under the close supervision of two experienced clinicians (Cynthia Kar Yung Yiu and Prasanna Neelakantan). 

### 2.2. Experimental Design

The experimental study design for the present study is shown in Figure 1. This cross-sectional study was conducted on 80 healthy children aged between 48 and 72 months old, 40 children with severe early childhood caries, and 40 caries-free children. The children were then further divided into four groups, with 20 children in each group: Group 1: S-ECC—*C. albicans* (*C. a*), Group 2: S-ECC—*M. globosa* (*M. g*), Group 3: CF—*C. albicans* (*C. a*), and Group 4: CF—*M. globosa* (*M. g*).

The anticipated presence proportion of *Candida* in the S-ECC lesions of children was estimated as 31.9% from a previous pilot study [9], and the sample size was calculated based on precision. To achieve a 95% confidence interval with a width of 20% (i.e., ±10%), a total of 84 children were required. However, due to the COVID-19 outbreak and suspension of dental services, only samples from 80 children could be obtained.

### 2.3. Inclusion Criteria

The American Academy of Paediatric Dentistry classification was used to identify children with S-ECC. The children selected for sample collection had at least two decayed (occlusal or proximal lesions) asymptomatic primary molars. The severity of the cavitated lesions was assessed by the same examiner using the well-established International Caries Diagnosis and Assessment System (ICDAS) classification. According to ICDAS, S-ECC was further sub-categorized as code 5 (a distinct cavity with visible dentine involving less than half the tooth surface) and code 6 (an extensive and distinct cavity with visible dentine affecting more than half of the surface). For this study, samples were only collected from decayed lesions coded as 5 and 6, as these are the most severe cavitated dentine caries lesions according to the ICDAS classification system [14].

### 2.4. Exclusion Criteria

The exclusion criteria for this study included children who had been on antibiotics or antifungals within the past 4 weeks, those who were wearing orthodontic appliances, children with any congenital tooth anomaly, and children whose teeth had a likelihood of pulp exposure during the caries removal process or were uncooperative during dental examination.

### 2.5. Caries Diagnosis

The caries status was evaluated using the WHO criterion of decayed, missing, and filled (dmft) tooth index, while the severity of cavitated lesions was determined as either caries code 5 or 6 according to ICDAS criteria [13], as described previously. Intra-examiner reproducibility at the tooth surface level was assessed using Cohen’s kappa coefficient (0.82). A single, trained, and calibrated operator collected infected dentine samples from 40 S-ECC children under a strict aseptic protocol.

### 2.6. Site-Specific Sample Collection

Three samples, saliva, plaque, and caries lesions, were collected from each participant with S-ECC. From each CF participant, two samples, saliva and plaque samples, were collected. The samples from caries lesions were obtained from symptom-free, caries-active, deep dentin lesions categorized as ICDAS codes 5 and 6. The teeth were cleaned using a prophy brush without the use of prophy paste and then dried. A sterile spoon excavator was used to collect the infected dentine samples from the occlusal and proximal lesions [9]. If gingival bleeding occurred during sample collection or if gingival bleeding contaminated the cavity, the samples were excluded. Immediately after collection, the samples from caries lesions were placed in a sterilized container containing PBS solution.

For the plaque samples, a supragingival dental plaque sample was collected from the buccal surfaces of maxillary incisors [15,16,17] using a sterilized cotton swab [17]. Samples were collected from clinically sound gingival areas for the CF group, and around the most cavitated enamel for the cervical and proximal caries group. Immediately after collection, the cotton swab was placed in a sterilized container containing PBS solution. Regarding the saliva samples, 2 mL of unstimulated whole saliva was collected from each child by having them spit directly into a sterile container. All samples were then transferred immediately to the laboratory for analysis.

### 2.7. Internal Transcribed Space (ITS) and Quantitative Real Time Polymerase Chain Reaction (PCR) 

The samples were initially centrifuged at 9880 rpm, and the resulting pellet was resuspended in 80 μL of 20 mM Tris-HCl, pH 8.0; 2 mM EDTA; 1.2% Triton with 20 μL of 20 mg/mL lysozyme and incubated overnight at 37 °C. The DNA was then extracted from the samples using the manufacturer’s protocol (QIAamp DNA Mini Kit, QIAGEN, Hilden, Germany). PCR analysis was conducted using the universal yeast primer sequences ITS4 and ITS86, as well as the universal primers for *M. globosa.* The load of the organisms was measured in real time using a Taqman TAMRA probe. Table 1 shows the sequence of primers for *C. albicans* and *M. globosa*. 

### 2.8. Statistical Analysis

The baseline data, including Ct values of *C. albicans* and *M. globosa* for the samples, were entered into a Microsoft Office Excel (Microsoft, Redmond, WA, USA) spreadsheet. For the analysis of *C. albicans* data, statistical tests such as Fisher’s exact test and binomial exact test were used. Fisher’s exact test was employed to compare the prevalence of *C. albicans* in saliva and plaque samples between the S-ECC and CF groups. A binomial exact 95% confidence interval was also provided to estimate the population prevalence (proportion of children having *C. albicans*) within the S-ECC group. For the analysis of *M. globosa* data, a two-sample *t*-test was performed when the results followed a normal distribution. However, if the results did not follow a normal distribution, the Mann–Whitney U test was performed. The statistical analysis was performed using SPSS Version 28.0 (SPSS Inc., Chicago, IL, USA). The results were considered statistically significant at *p* ≤ 0.05.

## 3. Results

### 3.1. Demographic Characteristics of the Participants

The study included a total of 80 children, consisting of 48 boys and 32 girls. Among them, 40 children had S-ECC, and 40 children had CF. All participants were asked about their history of antibiotics or antifungal treatment for any condition. The participants were healthy children aged between 48 and 72 months, with a mean age of 52 months. A total of 200 samples were collected, including saliva, plaque, and caries samples for each S-ECC participant (120 samples from the S-ECC group), and saliva and plaque samples for each CF child (80 samples from the CF group). Two samples (1 from the S-ECC group and 1 from the CF group for processing *C. albicans*) were excluded due to blood contamination.

### 3.2. Prevalence of Candida albicans

Real-time PCR (RT-PCR) analysis was performed to determine *Candida albicans* load, and the results are presented in Table 2, showing the log number of *C. albicans* in each sample. The PCR analysis clearly indicates that the load of *C. albicans* in saliva, plaque samples, and the samples from the caries lesions obtained from the S-ECC children was higher compared to the samples isolated from CF children. Intriguingly, the load of *C. albicans* was not detected in either the plaque or saliva samples from all CF children. This could be due to the *C. albicans* load in these samples being much lower than the detection threshold level. 

Table 3 represents the results of Fisher’s exact test, comparing the proportion of *C. albicans* in saliva and plaque samples between children with S-ECC and CF children. In the S-ECC group, the *C. albicans* proportion was 10.5% in saliva samples and 15.8% in plaque samples, whereas in the CF group, it was 0% in both samples. The differences in the *C. albicans* proportion between the S-ECC and CF groups for both saliva and plaque samples were found to be statistically significant (*p* < 0.05).

Table 4 shows the proportion of *C. albicans* in saliva and plaque samples in the S-ECC group. The proportion of *C. albicans* varied between 6 and 46% based on the binomial exact 95% confidence interval (Table 4).

### 3.3. Prevalence of M. globosa

Table 5 represents the cycle threshold values (Ct mean values) of *M. globosa*. The Ct values in all samples, both in the S-ECC and CF groups, were above 30, indicating a very low *M. globosa* load.

Table 6 compares the average cycle of threshold value of *M. globosa* in plaque and saliva samples between the S-ECC and CF groups. No significant difference was found in the mean load of *M. globosa* between the plaque samples of the S-ECC (39.5 *±* 1.3) and CF groups (40.0 *±* 0.9) (*p* = 0.159). Conversely, no significant differences were observed in the mean load of *M. globosa* between the saliva samples of the S-ECC group (38.3 *±* 1.4) and CF group (40.0 *±* 0.9) (*p* = 0.051). 

## 4. Discussion

Fungi, specifically *Candida*, are commensal microorganisms that can colonize dental surfaces [18]. *C. albicans* is the most frequently found *Candida* species in the oral cavity [9]. Our PCR analysis revealed a significantly higher load of *C. albicans* in saliva, plaque, and caries lesion samples from S-ECC children compared to samples from CF children. This finding is consistent with previous studies. Sziegoleit et al. [19] reported a higher frequency of *Candida* species in saliva samples from individuals with active caries (67%) compared to CF individuals (2%). 

Similarly, Akdeniz et al. [20] found a statistically significant difference in *Candida* prevalence between caries and CF children. Our study further supports the association of *C. albicans* with dental caries, as it was exclusively identified in children with S-ECC. These findings are in line with the growing evidence suggesting the potential role of *C. albicans* in caries development in preschool children. Our results also showed a higher proportion of *C. albicans* in the S-ECC group compared to the CF group in both saliva and plaque samples. However, no significant difference was observed in the prevalence of *M. globosa* between the plaque and saliva samples of the S-ECC and CF groups. Therefore, the null hypothesis that there is no difference in the prevalence of *C. albicans* and *M. globosa* in S-ECC children compared to CF children has to be partially rejected. Reducing *C. albicans* colonization can be an effective approach in reducing caries risk. Further investigations are warranted.

A series of seminal studies [8,21,22,23,24] have demonstrated that *Candida* species possess biofilm-forming abilities, acidogenic and aciduric potential, and other enzyme-mediated virulence attributes, all of which could contribute to cariogenicity. These fungi can survive in extremely low pH environments and produce enamel-dissolving short-chain carboxylic acids, as well as several protein-degrading hydrolases. The production of these acids leads to demineralization of enamel and dentine, while hydrolases (such as hemolysins, phospholipases, DNAses, acidic hydrolases) have the potential to degrade the organic components of dentine. 

Within the S-ECC group, we observed a wide range in the proportion of children with *C. albicans*, varying between 6 and 46%. This high variability can be attributed to differences in the dmft scores of the children. The presence of *C. albicans* may contribute to a lower maintenance of salivary pH, which can lead to the initiation of dental caries [25]. This finding is consistent with previous studies that reported a correlation between the isolation of yeast and an increased number of caries lesions [15,26,27]. The severity of caries may be attributed to the higher acidogenic potential of *C. albicans* [17]. A decrease in pH levels in the oral cavity creates a favorable environment for the activity of extracellular enzymes, which serve as important virulence factors for *Candida* [28]. Ten Cate et al. [29] further reported that *C. albicans* not only has the ability to adhere to hydroxyapatite (HAP), but they also dissolve HAP to a greater extent (approximately 20-fold) compared to *S. mutans*. In addition to adhering to tooth surfaces, *C. albicans* also exhibits high collagenolytic activity, denaturing collagen in exposed dentine [16].

Intriguingly, the prevalence of *C. albicans* was not detected in any of the saliva or plaque samples from CF patients. It is possible that the load of *C. albicans* in caries-free children was too low to be detected by RT-PCR analysis. This could be due to factors such as good oral hygiene practices. Thomas et al. [17] reported that *Candida* carriage ranges from 7 to 21% in CF children. Our finding contradicts the study by Thomas et al. [16], which demonstrated 100% *Candida* prevalence in saliva samples of caries-free children. However, our findings are consistent with the study by Thaweboon et al. [30], which reported an absence of *Candida* in the CF group. A future longitudinal study that follows a cohort of children over time to investigate the development of the oral microbiome and the prevalence of *C. albicans* in children with and without early childhood caries would provide a more comprehensive understanding of the relationship between *C. albicans* and early childhood caries.

Apart from *C. albicans,* other fungi are also commonly found in dental plaque in the oral cavities of children suffering from S-ECC [15]. These fungi are believed to contribute to caries progression by facilitating carbohydrate fermentation and producing acids. One such fungus is *Malassezia,* which is a lipid-dependent yeast known to be highly prevalent and abundant [31]. Interestingly, this study revealed that *M. globosa* was present in both the S-ECC and CF groups, suggesting that its presence alone may not be a definitive indicator of caries development. Baraniya et al. [7] reported a significant association between *M. globosa* and CF children. This could be attributed to the inhibitory effect of *M. globosa* against *S. mutans* through the secretion of antibacterial substances. Moreover, *M. globosa* can also inhibit biofilm formation by *Staphylococcus aureus* through a secreted aspartyl protease [32]. This suggests that *M. globosa* may play a role in preventing caries development by inhibiting the growth of cariogenic bacteria and preventing the formation of biofilms. Further research is needed to fully understand the role of *M. globosa* and other fungi in dental plaque and caries progression. In future studies, increasing the sample size and incorporating different investigation methods, such as metagenomic analysis and next-generation sequencing, can offer a more comprehensive understanding of the microbial dynamics involved in the relationship between the *Candida* biome and S-ECC.

There is compelling evidence to suggest that S-ECC has a detrimental impact on the quality of life of affected children, making it a significant public health concern [3,33]. These prevalence findings highlight the necessity for the development of more effective evidence-based strategies for preventing S-ECC [4]. However, without a clear understanding of the underlying causes of the disease, it is challenging to provide appropriate care to alleviate the condition and to design suitable care programs.

This study has a few notable limitations. First, the small sample size and recruitment of preschool children only from the Multi-Specialty Clinic limit the representativeness of the study to the wider population of children in Hong Kong. Additionally, the study’s focus on children aged 48–72 months may not capture the full spectrum of severe early childhood caries. To address these biases, future studies could recruit participants from multiple clinics or regions in Hong Kong, increase the sample size, and include a wider age range of children with severe early childhood caries. This would enhance the study’s generalizability. Second, the exact quantity of *M. globosa* could not be determined due to the absence of standard DNA samples. Thirdly, the study did not investigate the proportion of *C. albicans* and *M. globosa* or their association with the severity of the caries condition. Lastly, in vivo research was conducted on young children, which posed challenges related to compliance and obtaining parental consent. These factors hinder proper clinical examination and sample collection.

## 5. Conclusions

With major limitations, the study found a higher association between the prevalence of *C. albicans* and S-ECC compared to the caries-free group. Although present at low levels, *M. globosa* was found to be present in both caries-free and S-ECC groups. The results support the potential role of the oral mycobiome, primarily *Candida* species, in dental caries. Further research is needed to investigate the cariogenic attributes and associated virulence profiles of oral yeast isolates from S-ECC lesions of preschool children and to explore the inter-kingdom correlations between *C. albicans* and *M*. *globosa*.

## Figures and Tables

**Figure 1 healthcare-12-01359-f001:**
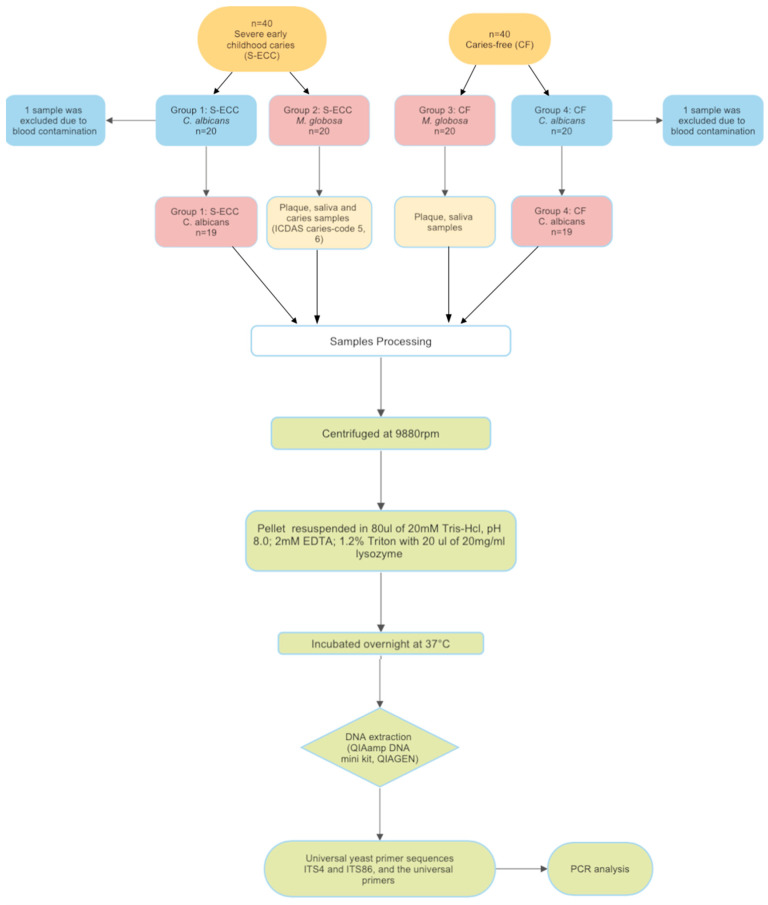
Flowchart of the experimental design.

**Table 1 healthcare-12-01359-t001:** Sequence of primers for *C. albicans* and *M. globosa*.

Organism		Sequence
*C. albicans*	F	5′-GTG AAT CAT CGA ATC TTT GAA C-3′
	R	5′-TCC TCC GCT TAT TGA TAT GC-3′
	Probe	FAM−ATT GCT TGC GGC GGT AAC GTC C−TAMRA
*M. globosa*	F	5′-GGCCAAGCGCGCTCT-3′
	R	5′-CCACAACCAAATGCTCTCCTACAG-3′
	Probe	5′-FAM-ATCATCAGGCATAGCATG-BHQ1

**Table 2 healthcare-12-01359-t002:** The load of *C. albicans* from samples isolated from patients.

	Severe Early Childhood Caries Patients			Caries-Free Patients	
Patients	Saliva	Plaque	Caries	Saliva	Plaque
1	4.26 × 10^5^	2.98 × 10^4^	2.06 × 10^7^	0	0
2	0	1.23 × 10^4^	3.61 × 10^1^	0	0
3	3.37 × 10^5^	9.60 × 10^4^	0	0	0
4	8.41 × 10^4^	2.50 × 10^4^	0	0	0
5	3.31 × 10^4^	1.61 × 10^3^	3.63 × 10^5^	0	0
6	5.28 × 10^4^	2.92 × 10^4^	7.89 × 10^5^	0	0
7	9.81 × 10^5^	1.25 × 10^8^	2.41 × 10^8^	0	0
8	2.26 × 10^7^	6.65 × 10^7^	3.87 × 10^8^	0	0
9	9.81 × 10^5^	1.25 × 10^8^	2.41 × 10^8^	0	0
10	2.26 × 10^7^	6.65 × 10^7^	3.87 × 10^8^	0	0
11	4.17 × 10^7^	2.60 × 10^8^	4.55 × 10^7^	0	0
12	4.60 × 10^6^	2.22 × 10^8^	5.50 × 10^8^	0	0
13	5.17 × 10^9^	0.00 × 10^0^	0.00 × 10^0^	0	0
14	2.14 × 10^6^	3.00 × 10^8^	1.79 × 10^9^	0	0
15	3.48 × 10^7^	1.83 × 10^8^	1.49 × 10^8^	0	0
16	3.43 × 10^9^	0.00 × 10^0^	0.00 × 10^0^	0	0
17	7.88 × 10^8^	0.00 × 10^0^	7.88 × 10^8^	0	0
18	0.00 × 10^0^	2.45 × 10^4^	4.02 × 10^5^	0	0
19	3.94 × 10^6^	4.58 × 10^6^	3.33 × 10^7^	0	0
Total	5.01 × 10^8^	7.13 × 10^7^	2.44 × 10^8^	0	0

**Table 3 healthcare-12-01359-t003:** Comparison of *C. albicans* in saliva and plaque samples between severe early childhood caries and caries-free groups.

	Severe Early Childhood Caries Group (*n* = 19)	Caries-Free Group (*n* = 19)	*p*-Value
Saliva	2 out of 19 (10.5%)	0 out of 19 (0%)	<0.001 *
Plaque	3 out of 19 (15.8%)	0 out of 19 (0%)	<0.001 *

* Fisher’s exact test.

**Table 4 healthcare-12-01359-t004:** The proportion of children with *C. albicans* in the severe early childhood caries group.

Variable	Obs	Proportion	Std. Err.	Binomial Exact [95% Conf. Interval]	
	19	0.2105263	0.0935288	0.0605245	0.4556531

cii proportions 19 4, exact.

**Table 5 healthcare-12-01359-t005:** The average cycle of threshold value of *M. globosa* from samples isolated from patients.

Patients	Severe Early Childhood Caries Patients			Caries-Free Patients	
	Saliva	Plaque	Caries	Saliva	Plaque
1	38.19791	39.46937	40.01772	39.54087	38.90553
2	33.5537	40.00582	41.23515	38.46895	41.3937
3	37.70129	39.24055	36.87447	31.65252	40.22517
4	38.06826	38.77831	39.71342	31.30251	40.41551
5	39.55402	39.84827	39.83411	36.12486	40.6979
6	39.79036	41.74977	40.66396	32.87978	39.61966
7	37.18097	39.62401	40.67602	38.10505	39.95837
8	40.27142	41.22569	40.15009	33.39236	40.07108
9	38.86098	38.22488	41.96599	31.18872	39.05045
10	37.80894	39.75838	40.27203	39.42402	38.85844
11	39.39632	38.02472	38.64331	38.52523	40.55655
12	38.53099	37.39825	38.54734	40.95286	41.88996
13	37.68338	37.4791	38.86926	36.91849	41.30675
14	37.3256	37.28071	38.17343	37.98817	39.46434
15	38.43781	41.17571	37.61873	38.70189	40.42744
16	38.37862	40.84421	39.95787	39.15549	39.95975
17	38.76104	39.95461	41.43936	36.90451	39.86407
18	39.01728	40.1673	39.82533	30.79726	38.9537
19	38.76104	39.95461	41.43936	37.63364	40.15736
20	38.76104	39.95461	41.43936	38.22379	38.53745

**Table 6 healthcare-12-01359-t006:** Comparison of the average cycle of threshold value of *M. globosa* in saliva and plaque samples between severe early childhood caries and caries-free groups.

	Severe Early Childhood Caries Group (*n* = 20)			Caries-Free Group (*n* = 20)			Comparison between 2 Groups (*p*-Value)
	Mean	Std. Deviation	Median	Mean	Std. Deviation	Median	
Saliva	38.3020	1.38032	38.4844	36.3940	3.25101	37.8109	0.051 #
Plaque	39.5079	1.29769	39.8033	40.0157	0.90436	40.0154	0.159 *

# Mann-Whitney test; * Two sample *t*-test.

## Data Availability

The data that support the findings of this study are available from Cynthia KY YIU upon reasonable request.

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
