# Peer review of "The Prevalence of Candida albicans and Malassezia globosa in Preschool Children with Severe Early Childhood Caries: A Case-Control Study"

_healthcare, 2024, doi:10.3390/healthcare12131359_

Round 1

Reviewer 1 Report

Comments and Suggestions for Authors

The authors developed an evaluation comparing the loads of Candida albicans and Malassezia globosa in caries-free individuals and those with severe early childhood caries. I have some questions and recommendations, especially regarding the Material and Methods section.

Overall question:

It is not clear to me why the authors designed the groups according to the microorganisms and the caries status rather than considering only the caries status. It would be more reasonable to recrute caries-free and S-ECC children and assess the loads of each microorganism. Would not it? Please, clarify this aspect.

I suggest that Malassezia globosa to be also included to the title as it was also assessed in the work. The title should be restructured considering it. Also, severe early childhood caries should not be abbreviated in the title. Finally, Candida-biome seems too much comprehensive as the authors  evaluated only Candida albicans.

In the abstract, all abbreviations should be written in full when first time mentioned.

Introduction:

Please revise this section regarding spelling errors.

Material and Methods:

The authors should rethink the terminology 'caries samples'. Caries is the disease. You collected the samples from the lesions. The authors should use 'caries lesions' or maybe 'cavitation'.

Discussion

The authors should be careful when stating 'This study is the first pilot study to assess the Candida-biome of S-ECC in Hong Kong  children'. The sample does not seem representative for all the Hong Kong children.

Tables 

Please improve the quality of the tables in terms of captions. All abbreviations should be written in full in the captions.

Comments on the Quality of English Language

The manuscript could substantially benefit from English language and scientific writing revision.

Reviewer 2 Report

Comments and Suggestions for Authors

The study presented to me for evaluation is an interesting proposition for the reader and another interesting voice in the discussion about the possible involvement of Candida spp. in the development of dental caries.

As a reviewer, I propose the following changes:

Abstract

L-10  SECC -S-ECC

L-14 expand shortcuts ITS, PCR, ICDAS

Introduction

L-38 [6,7,8,]   - {6, 7, 8]

L-42-53 I would add a short paragraph about non-albicans Candida and its involvement 

M&M 

L-85 expand shortcuts CKY and PN

L-89 C. albicans (CA) - C. albicans (C. a);

         M. globosa (MG)- M. globosa (M. g)

L-91-95 It's great that the number of participants in the study was statistically calculated

L -101 at least 2 two - at least 2 (two)

L-111-114 in the exclusion criteria I would add not only antibiotics but also antifungal drugs (in 48-72 months old oral candidiasis is common)

L-137 2mL - 2 mL

L-143-144 the same sytuation 80uL, 20mM, 2mM, 20mg/ml - 80 uL, 20 mM, 2 mM, 20 mg/mL

Results are well presented

Discussion

L- 261 7%-21%  - 7-21%

Conclusion 

In conclusion, this study provides ....-  This study provides

Reviewer 3 Report

Comments and Suggestions for Authors

This is a very meaningful paper that studied the correlation between S-ECC and Candida biome. There were previous studies that ECC was associated with C. albicans, and the fact that it was detected by real-time PCR and compared with the caries free group is also reliable in terms of precision. However, it is somewhat regrettable that the research results on M.globosa were found that there was no statistically significant difference from S-ECC unlike previous research results, leaving more need for future research. In addition, when C.albicans was present, it would have been better if there had been consideration or inference about whether it was clinically significant to control it and whether it was increasing the caries risk. Since S-ECC cannot be regarded as linearly increasing the severity of disease compared to ECC due to the definition of S-ECC, the interpretation of what the quantified value of C. albicans means can be carried out as a future research project. It would be great if these were added to the discussion part. Still, it was difficult to conduct in vivo research on children under the age of 6. There is room for further research.

Round 2

Reviewer 1 Report

Comments and Suggestions for Authors

Overall, the authors improved the overall quality of their work by performing the changes suggested.

However, I am still concerned about the grouping. I have not understand yet why the authors did not evaluate all individuals CF and SECC for both the microorganisms assessed. It sounds a bit unreasonable. Assessing both microorganisms in CF individuals and SECC would increase the sample size and sounds a more reasonable design.

Also, the authors stated that "Reducing C. albicans colonization can be an effective approach in reducing caries risk. This can be achieved through using antifungal agents or probiotics. Further investigations are warranted." 
Even though the authors stated that further investigations are warranted, using antifungals for reducing C albicans in order to reduce caries is unreasoble considering caries aetiology and the side effects of this medication. This should be reconsidered.
